# TOOL CALLING: ENHANCING MEDICATION CONSULTATION VIA RETRIEVAL-AUGMENTED LARGE LANGUAGE MODELS

## ABSTRACT

Large-scale language models (LLMs) have achieved remarkable success across various language tasks but suffer from hallucinations and temporal misalignment. To mitigate these shortcomings, Retrieval-augmented generation (RAG) has been utilized to provide external knowledge to facilitate the answer generation. However, applying such models to the medical domain faces several challenges due to the lack of domain-specific knowledge and the intricacy of real-world scenarios. In this study, we explore LLMs with RAG framework for knowledge-intensive tasks in the medical field. To evaluate the capabilities of LLMs, we introduce MedicineQA, a multi-round dialogue benchmark that simulates the real-world medication consultation scenario and requires LLMs to answer with retrieved evidence from the medicine database. MedicineQA contains 300 multi-round question-answering pairs, each embedded within a detailed dialogue history, highlighting the challenge posed by this knowledge-intensive task to current LLMs. We further propose a new *Distill-Retrieve-Read* framework instead of the previous *Retrieve-then-Read*. Specifically, the distillation and retrieval process utilizes a tool calling mechanism to formulate search queries that emulate the keyword-based inquiries used by search engines. With experimental results, we show that our framework brings notable performance improvements and surpasses the previous counterparts in the evidence retrieval process in terms of evidence retrieval accuracy. This advancement underscores the framework's potential to effectively address the inherent challenges of applying RAG models to the medical domain.

## 1 INTRODUCTION

Large language models (LLMs) (Achiam et al., 2023; Touvron et al., 2023; Team et al., 2023) have revolutionized the field of natural language processing, showing remarkable impacts with the well-documented emergence of zero-shot capabilities in a variety of downstream tasks, like machine translation (Zhang et al., 2023c), text generation (Kojima et al., 2022) and machine reading comprehension (Samuel et al., 2023). Such impressive abilities stem from the ever-increasing number of parameters and large-scale training corpus.

Despite the massive knowledge, LLMs still struggle with considering issues of hallucination (i.e., prone to generate factually incorrect statements) (Bang et al., 2023; Ji et al., 2023) and temporal misalignment (i.e., unable to capture the changing world) (Kandpal et al., 2023) in a set of tasks (Yin et al., 2022; Lewis et al., 2020). Such knowledge-intensive tasks require access to a vast amount of knowledge beyond the training data, hindering wider practical applications of LLMs since further validation of responses needs to be conducted. Towards this issue, existing methods (Li et al., 2023b; Jiang et al., 2023; Xu et al., 2023; Wang et al., 2023; Cheng et al., 2024) incorporated external knowledge with LLMs by retrieval augmentation, dubbed as Retrieval Augmented Generation (RAG). In detail, LLMs retrieve the relevant information for the input query and utilize the retrieved evidence as additional context to generate the response. Such *Retrieve-then-Read* framework cleverly combines flexible knowledge sources in a non-parameterized form for knowledge-intensive tasks and has become one of the hottest paradigms to alleviate the drawbacks in naive LLM generations.

Figure 1: The medication consultation: a detailed discussion between healthcare professionals and users about prescribed medications, including their names, indications, usage, side effects, etc. Professionals utilize the knowledge in the medicine database to provide a more robust response.

With recent advancements, LLMs hold great promise for facilitating specific domains like medical fields (Li et al., 2023c; Singhal et al., 2023; Li et al., 2023a; Zhang et al., 2023a; Xiong et al., 2023). Beneath the advancements, we find a notable gap in applying LLMs to medical fields, especially for knowledge-intensive tasks like medication consultation. As shown in Figure 1, medication consultation aims at providing real-time accessibility for medication-related inquiries and enhancing medication safety through searching from the database, requiring depth in domain-specific areas. The dialogs in real-world scenarios are usually ambiguous and verbose, e.g., users tend to use layman's terms instead of standard terms and provide much more information than what might be medically relevant. This poses a challenge to retrieve appropriate evidence from the medicine database based on user input. Moreover, attempts to assess the capabilities of RAG-based LLMs in medical scenarios are limited. Based on these premises, we ask: *Is the LLM with vanilla RAG enough for the medication consultation?*

To evaluate the proficiency of LLMs vanilla RAG in medication consultation scenarios, we introduce MedicineQA, a benchmark with a medicine database serving as the knowledge. We recruited a panel of 5 board-certified physicians to create the benchmark as follows: sourcing and rephrasing questions from an online medical consultation website, simulating multiple rounds of dialogue scenarios, and retrieving and determining reference evidence. Consequently, MedicineQA contains 300 samples, covering most medicines commonly used in real-world scenarios across ten aspects of medicine application. Considering how to retrieve appropriate evidence from the database based on user input is crucial for LLMs with RAG. In the MedicineQA, we provide reference evidence for each sample, supporting the evaluation of the retrieval process. To the best of our knowledge, MedicineQA, along with its medicine database, is the first benchmark in the medical domain to evaluate the accuracy of the retrieval process. Our further experiments reveal that vanilla RAG methods suffer from serious challenges in retrieving relevant information with intricate dialogue history.

To generate a simple yet robust search query from intricate dialogue history, we propose RagPULSE based on PULSE Zhang et al. (2023b). Instead of the *Retrieve-then-Read* framework adopted by previous retrieval-augmented work, RagPULSE utilizes a novel *Distill-Retrieve-Read* framework to access the external knowledge. Specifically, we prompt RagPULSE to summarize the medication inquiry and the dialogue history to keywords for several predefined search engines, mimicking how a human would use search engines. The RagPULSE integrates the evidence retrieved from the medicine database to formulate a comprehensive response. By training on the synthetic dataset for "tool calling," RagPULSE demonstrates strong capabilities in generating accurate queries and achieves remarkable performance in dealing with medication consultation. Our main contributions can be summarized as follows:

- We present MedicineQA, a benchmark comprising 300 high-quality, expert-annotated multi-round dialogues spanning ten key aspects of medication consultation that users commonly encounter on online consultation platforms.
- We propose a pioneering retrieval augmentation framework, *Distill-Retrieve-Read*, to generate robust query from intricate dialogue history via the "tool calling" mechanism.
- Incorporated with the framework, our proposed RagPULSE outperforms all publicly available models in performance and is competitive with state-of-the-art commercial products with a smaller parameter size.

## 2 RELATED WORK

**Large Language Model in Medical Domain.** The impressive abilities of large language models (LLMs) across various applications have catalyzed extensive investigation into employing them in healthcare and medical domains. This surge in attention is documented through a growing body of research (Thirunavukarasu et al., 2023; Clusmann et al., 2023). Some recent works have studied to augment LMMs with real-world data. ChatDoctor (Li et al., 2023c), trained by fine-tuning LLaMA (Touvron et al., 2023) on a large dataset of patient-doctor dialogues, achieves high accuracy and reliability in medical scenarios with an external information retrieval module. From the other line, some adopt the synthetic data for fine-tuning. Zhang et al. (2023a) utilized real-world data from medical professionals alongside distilled data from ChatGPT to fine-tune the model. To enhance the capability in the multi-round conversation, BianQue (Chen et al., 2023) trained the model on a self-constructed dataset containing multi-round inquiries and health suggestions. Despite the remarkable performance, there is still a gap in applying LLMs in real-world scenarios due to the lack of domain-specific knowledge. To further evaluate the proficiency of LLMs in medical domains, we introduce MedicineQA, a benchmark derived from real-world medication consultation scenarios.

**Retrieval-Augmented Generation.** LLMs require external knowledge to alleviate the factuality drawbacks. Retrieval-augmented generation (RAG) has been regarded as an effective solution to mitigate the aforementioned hallucinations and temporal misalignment issues inherent in large language models, especially for knowledge-intensive tasks. Generally, studies of RAG can be categorized into three types (Gao et al., 2023), namely Naive RAG, Advanced RAG, and Modular RAG. Naive RAG means a straightforward *Retrieve-then-Read* framework (Lewis et al., 2020; Karpukhin et al., 2020; Izacard et al., 2022). To enhance retrieval quality, the Advanced RAG builds upon the foundation of Naive RAG by incorporating pre-retrieval (Li et al., 2023b) and post-retrieval (Jiang et al., 2023; Xu et al., 2023) strategies. Modular RAG improves the overall performance by decomposing the *Retrieve-then-Read* framework into fine-grained modules with distinct functionalities, such as a search module(Wang et al., 2023), memory module(Cheng et al., 2024).

## 3 METHOD

In Section (3.1), we propose MedicinceQA, a novel benchmark to evaluate LLMs' capabilities toward knowledge-intensive tasks in medical fields. We curate the benchmark from various real-world medication consultation scenarios and unified them into multi-round dialogue. Then, we present RagPULSE in Section (3.2), a dedicated pipeline that adopts *Distill-Retrieve-Read* framework for multi-round medication consultation. The fundamental operations of RagPULSE comprise three main steps: (1) the LLM calls the search engine tool and distills the dialogue history into a new query to gather evidence from the external medicine database; (2) the generated search query is executed to retrieve related evidence following a hierarchical form; (3) the retrieved evidence is provided to the LLM, and the LLM respond the user's question by the retrieved evidence.

### 3.1 BENCHMARK CREATION

Existing benchmarks for evaluating the capabilities of LLMs in medical fields primarily focus on widely known or widely available tasks given a specific context (e.g., Automatic Structuring of medical reports and Named Entity Recognition). However, these benchmarks are insufficient for assessing LLMs' proficiency in knowledge-intensive tasks. Therefore, we introduce MedicineQA, a novel benchmark designed for evaluating LLMs within the context of medication consultation. Along with the medicine database, MedicineQA also provides ways to judge the robustness of generated search keywords and evaluate the accuracy of the retrieval process.

**Data Collection.** In an effort to align the benchmark with real-world scenarios, our dataset was compiled from several online consultation websites, commonly referred to as "internet hospitals," which comprise numerous online consultation records between users and medical experts. Specifically, we crawled data from five major online consultation websites following previous works [1]. These websites provide a rich source of anonymized patient-doctor dialogues, ensuring no risk of personal information leakage. Each record contains multiple rounds of dialogue, we categorized each

---

[1] https://mlpcp21.github.io/pages/challenge.html

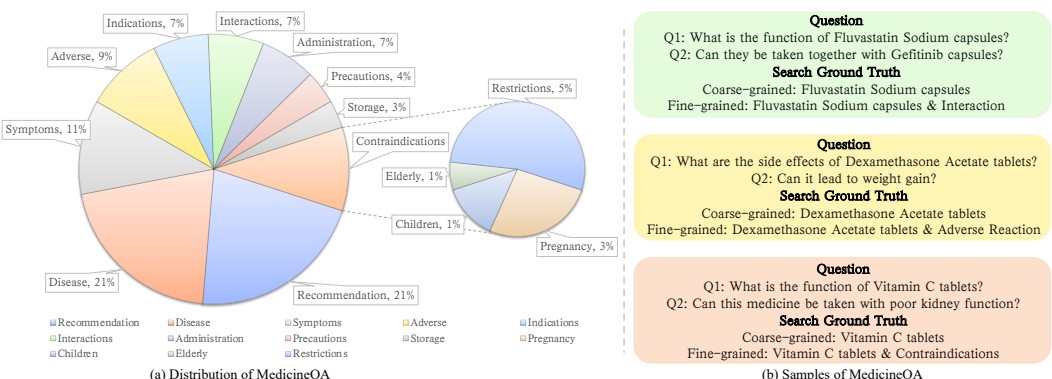

Figure 2: (a) The distribution of our proposed MedicineQA. MedicineQA involves ten specific scenarios of the medication consultation. The distribution of the benchmark is similar to that of the real scenario. (b) Samples of the benchmark: Interaction, Adverse reactions, and Contraindications. Our benchmark is available in both English and Chinese.

record into three categories: 1) Diagnostic Process, where the expert diagnoses based on symptoms provided by the user; 2) Medication Consultation, where the expert addresses queries regarding medications for certain conditions; 3) Other, which includes the patient's medical history and some trivial communication. In total, we amassed 1,028,090 records comprising 6.24M pairs.

**Data Refinement.** Given the crawled data, we first conducted an initial statistical analysis and identified the 200 most commonly mentioned medicines as the scope for further processing. To ensure the correctness, we recruited a panel of 5 board-certified physicians to curate the content. The physicians filtered out irrelevant dialogues of each selected record and summarized it into one question about a specific medicine. For each summarized question, we utilized GPT-4 Achiam et al. (2023) to expand them into multi-round dialogue according to the context of the relevant record. This approach ensured that the generated dialogue content accurately reflected real-world scenarios. To prevent GPT-4 from hallucinating inappropriate content, physicians manually revised the dialogues to ensure a logical progression of questions, with each answer building on the information provided in the preceding dialogues and without repeating information. As a result, MedicineQA consists of 300 samples covering over 150 medicines, spanning ten aspects (from Recommendation to Storage). More details can be seen in the Appendix A.1.

**Medicine Database.** To provide precise and structured information, we introduce an entity-oriented medicine database with 42,764 medicines, where each medicine is represented in three forms: brand name, generic name, and detailed attributes like usage, contraindications, adverse reactions, etc. The medicine database is a small subset of an authorized database. The full database contains detailed descriptions of approximately 192,000 medicines from a collaborated company with the authorization of a publishing house. Each medication document has undergone a rigorous triple review and verification process to ensure compliance with established medical standards. Formally, for each medicine $M_i$ in our database $D$, we first concatenated its generic name with each attribute $a_j$ to obtain the entity-attribute items $E_{ij}$, respectively. Then, each item is embedded into vectors and stored in a tree form according to the entity, i.e., the information of the medicine $M_i$ is stored in the form of $E_i = \{E_{i1}, E_{i2}, E_{i3}, \dots\}$, accompanied by its corresponding keys $K_i^n$ and $\{K_{i1}^a, K_{i2}^a, K_{i3}^a, \dots\}$. In our database $D$, $E_i$ and $E_{ij}$ can be obtained via $D[K_i^n]$ and $D[K_{ij}^a]$, respectively.

**Annotation.** In our benchmark, each question is associated with the corresponding medicine descriptions extracted from the medicine database, to serve as the retrieved evidence. The detailed process of constructing the evidence can be found in Appendix A.2. To evaluate the retrieval process, we further labeled two types of retrieval ground truths: one is the document-level for coarse-grained evaluation $K_c$, and the other is the specific sections in the relevant documents for fine-grained attribute-level assessment $K_f$. One sample of our MedicineQA can be formulated as $\mathbf{S} = \langle H, Q_{T+1}, K_c, K_f \rangle$, where $H = \{(Q_i, A_i)\}, i = 1, 2, \dots, T$ is the dialogue history, $(Q_i, A_i)$ denotes a round of conversation between the user and the agent, and $T$ is the number of

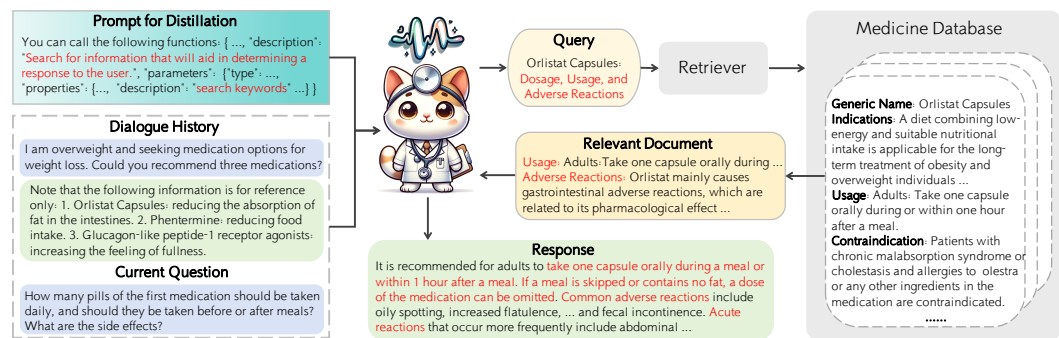

Figure 3: An example of how RagPULSE deals with user inquiry about the usage and adverse reactions of Orlistat Capsules in the daily medication consultation scenario. The "Prompt for Distillation" serves as the system prompt within the *Distill-Retrieve-Read* framework, indicating that various search engines are available for information retrieval. The overall workflow of RagPULSE consists of three steps: (1) Distilling the key information and forming the searching query from the combination of dialogue history and current Question; (2) Retrieving the corresponding medicine evidence from the medicine database via the generated search query; (3) Generating the response according to the retrieved evidence.

dialogue rounds. $Q_{T+1}$ represents a question about one specific medicine. $K_c, K_f$ are the coarse-grained and fine-grained ground truth for evaluating the retrieval process, respectively. In detail, $K_c$ is the $K_i^n$ in $D$, and $K_f$ is a subset of $\{K_{i1}^a, K_{i2}^a, K_{i3}^a, \dots\}$. We display the relative distribution of our proposed benchmark and present samples of the created data in Figure 2.

## 3.2 RAGPULSE

We choose PULSE (Zhang et al., 2023b) as the LLM, which demonstrates impressive performance in the medical field, and augment it with the *Distill-Retrieve-Read* framework. As shown in Figure 3, the process can be formulated into three steps. The LLM is first tasked to call the search engine tool and summarize the search query supported by the combination $[H, Q_{T+1}]$. Subsequently, the search engine retrieves relevant keys $\hat{K}$ from the medicine database $D$ and obtains the evidence $\hat{E}$ from the medicine database $D$. Finally, the LLM generates the answer $A_{T+1}$ according to $[H, Q_{T+1}, \hat{E}]$.

**Tool Calling.** How to retrieve appropriate evidence from the medicine database based on user input is crucial. The correctness and completeness of the search query directly impact the accuracy of the retrieval process. A simple but robust retrieval query is vital to clarify the search need from the context and eliminate irrelevant information in the external knowledge base. Recent studies either directly adopt the query from the dataset (Liu et al., 2024) or rewrite it by the black-box generation (Ma et al., 2023). However, there is inevitably a gap between the query and the evidence that needs to be obtained, especially for such a task with a long context. Only relying on the original capability of the LLM and human-written prompt lines makes it difficult to summarize correct inquiries from the intricate context while preserving key information. Inspired by the *program of thought (PoT)* (Chen et al., 2022), where the LLM generates Python code for retrieving, we integrate "tool calling" with the LLM. Specifically, we predefine several search engines in the system prompt for the LLM and instruct the LLM that it can retrieve useful information by generating a search query and then retrieving the necessary data via the specified search engine. This approach prompts the LLM to generate search keywords for search tools, mimicking the use of search engines. With the above paradigm, the LLM is able to call the search tool and generate the retrieval query according to the current dialogue.

**Synthetic Dataset.** To endow the LLM with the distillation ability, we construct a synthetic dataset for the dialogue distilling task following previous works (Ma et al., 2023; Hsieh et al., 2023; Ho et al., 2022). First, we collect a large-scale question set (including but not limited to dialogue questions and search engine questions) from several websites (e.g., Google and Baidu). Then, the selected questions are distilled and summarized as pseudo labels by prompting GPT-4 (Achiam et al., 2023)

| The Template of instructions for Tool Calling | Samples of Synthetic Data |
|---|---|
| ```
You can call the following tools:
{ "name": "search_engine",
 "description": "Search for information that
 will help determine a response to the user.",
 "parameters":
    {"type": "object",
    "properties": {"input": {"type": "string",
    "description": "search keywords"}},
    "required": ["input"]} }
``` | **Input**: 2017 college entrance examination ticket, fully opened, how much longer? How wide is it? 
**Output**: search_engine(2017 College entrance examination ticket size.) 

**Input**: How much does it cost for high school students to study in Japan? 
**Output**: search_engine(The cost of studying in Japan high school.) 

**Input**: When is there a typhoon in Guangzhou? 
**Output**: search_engine(Guangzhou Typhoon Forecast.) |

Table 1: The instructions and samples of the synthetic dataset for fine-tuning the LLM.

to utilize function call. As a result, we obtained 161,100 samples to prompt LLMs to distill the context into search keywords for the predefined search engines. After fine-tuning, the LLM shows remarkable performance in distilling the context into simple inquiries containing key information. The samples of synthetic data and the instructions for "tool calling" are shown in Table 1.

## 4 EXPERIMENTS

In this section, we measure the performance of RagPULSE on MedicineQA and compare it to existing LLMs and commercial products (4.2). We ablate the *Distill-Retrieve-Read* on the MedicineQA dataset, showing their importance (4.3). Finally, we present some cases to investigate the hallucinations of LLMs towards medication consultation.

### 4.1 EXPERIMENTAL SETTINGS

**Implementation Details** We develop RagPULSE with *Distill-Retrieve-Read* framework in Pytorch (Paszke et al., 2019) and fine-tune it by the proposed synthetic dataset. To enable PULSE to perform dialogue distillation while maintaining the capabilities in medical domains, we add the synthetic dataset to the fine-tuning datasets of PULSE. It is worth noting that a single machine with eight NVIDIA A100 GPUs proved sufficient for the memory requirements of PULSE (Zhang et al., 2023b). Our training framework integrates tensor parallelism (Wang et al., 2022) and ZeRO-powered data parallelism (Rajbhandari et al., 2020). We utilize the Adam optimizer with a weight decay setting of $0.01$ and betas of $(0.9, 0.95)$. The learning rate gradually decays from $9 \times 10^{-6}$ to $9 \times 10^{-7}$ following a cosine annealing learning rate schedule. To further accelerate training without sacrificing accuracy, we implement mixed-precision training, where we execute forward and backward computations in BFloat16 and conduct optimizer updating in Float32. For the compared models, we adopt the pre-trained weights and settings provided on the official website.

**Baselines.** Given the variety of current LLMs and the fact that MedicineQA is the medical domain, we choose open-sourced models and commercial products with notable performance in the medical domain to fully explore the current proficiency of LLMs in medication consultation scenarios. For a fair comparison, we utilize models that the results can be reproduced as follows: Doctor-GLM (Xiong et al., 2023), ChatGLM3 (Du et al., 2022), BianQue2 (Chen et al., 2023), MING (Liao et al., 2023), QWen2 (Bai et al., 2023), Baichuan2 (Baichuan, 2023) and GPT-3.5. We first prompt them to summarize the *Dialogue History H* and *Current Question $Q_{T+1}$* into a search query using the instruction (*Based on the above conversation about medical inquiries and medication queries, please summarize the search keywords for the user's final question using the dialogue record. Retrieve relevant medication information and return it in JSON format as follows:*{*"query": ...*}) The generated search query is then used to query the database for retrieving evidence. The HR@K can be calculated for the baseline models according to the *Relevant Evidence*.

**Metrics.** To evaluate the accuracy of the evidence retrieval stage, we employ the Hit Rate (HR@num), which represents the proportion of instances where the retrieval candidates contain the corresponding knowledge, with "num" indicating the number of candidates to be retrieved. We respectively calculate the hit rate of coarse-grained and fine-grained retrieval through the retrieved database key and the search ground truth. It should be noted that there are multiple ground-truth evidence entries for the aspect of Medication Recommendation. We adopt a strict evaluation metric: Assume the number of retrieved evidence $E$ is $x$ and the number of ground truth $G$ is $y$. If $x \le y$, re-

| Model Name | Param. Size | Ins. follow rate (%) | Retrieved Doc. (%) | | | Retrieved Attr. (%) | | | Generation | |
|---|---|---|---|---|---|---|---|---|---|---|
| | | | HR@1 | HR@5 | HR@10 | HR@1 | HR@5 | HR@10 | Elo Rating | Elo Rank |
| BianQue2 | 6B | 3.33 | 7.33 | 9.00 | 10.00 | 1.67 | 2.00 | 2.00 | 862 | 12 |
| DoctorGLM | 6B | 47.00 | 12.67 | 15.00 | 16.00 | 2.33 | 2.67 | 3.00 | 896 | 11 |
| ChatGLM3 | 6B | 92.33 | 27.33 | 32.00 | 34.00 | 8.00 | 9.33 | 9.67 | 979 | 9 |
| MING | 7B | 8.00 | 20.00 | 28.33 | 30.67 | 5.67 | 7.67 | 8.00 | 1002 | 7 |
| BenTsao | 7B | 16.67 | 33.33 | 45.33 | 48.00 | 12.67 | 17.33 | 18.33 | 889 | 11 |
| Baichuan2 | 14B | 98.33 | 52.67 | 66.67 | 71.33 | 26.67 | 35.33 | 38.00 | 1037 | 6 |
| QWen2 | 14B | 100.00 | 57.67 | 68.33 | 76.67 | 25.33 | 28.33 | 30.33 | 998 | 8 |
| GPT-3.5 | - | 100.00 | 63.67 | 72.33 | 78.67 | 27.00 | 31.33 | 32.67 | 1068 | 3 |
| GPT-4 | - | 100.00 | 62.33 | 76.33 | 82.00 | 26.67 | 32.33 | 34.00 | - | - |
| RagPULSE | 7B | 100.00 | 63.67 | 73.00 | 78.33 | **28.33** | **32.00** | **33.33** | 1060 | 4 |
| PULSE | 20B | - | - | - | - | - | - | - | 1041 | 5 |
| RagPULSE | 20B | 100.00 | **65.67** | **75.33** | 78.33 | 27.33 | 31.67 | 32.33 | 1074 | 2 |
| PULSE* | 20B | - | - | - | - | - | - | - | **1094** | 1 |

Table 2: Evaluation on MedicineQA. Our study employs the PULSE model with varying parameter sizes, augmented by the *Distill-Retrieve-Read* framework. We compare them with other LLMs and commercial products. "Retrieved Doc." refers to the process of only searching the generic name of the medicine (coarse-grained), while "Retrieved Attr." denotes calculating the results via the combination of the generic name and the specific attribute (fine-grained).

trieval is considered successful only when $E \subseteq G$. If $y \leq x$, retrieval is considered successful only when $G \subseteq E$. Given the answer of the medication consultation is in the form of free text, which is a challenge for evaluating the correctness, we utilize the Elo rating system (Elo, 1967; Chiang et al., 2023; Dettmers et al., 2023) to gauge the performance of LLMs on MedicineQA. It adjusts a player's rating based on the outcome of their games, taking into account the expected score versus the actual score. In our settings, each model is one competitor, and the powerful GPT-4 (Achiam et al., 2023) serves as the referee to determine which model performs better. More details can be seen in the Appendix A.3.

## 4.2 RESULTS

Here we thoroughly evaluate models using the MedicineQA benchmark. To assess the performance of evidence retrieval, we prompt those baseline models to formulate search queries by summarizing preceding dialogues and then calculate their accuracy in retrieving relevant evidence. Due to the limitations of some baseline models in retrieving evidence from the medicine database, we immediately adopt the attached corresponding medicine information as the context to guide the generation of the final responses. It is worth noting that our RagPULSE leverages the retrieved evidence to generate the answer. Experimental results are reported in Table 2.

We can see that some open-sourced models with smaller model sizes suffer from following the instructions for summarizing key information in specific format from complex dialogue histories, highlighting the inherent difficulties in medication consultation tasks. Finetuned on the synthetic dataset, our RagPULSE (7B) presents a surprising performance in the instruction following rate. This outcome validates the effectiveness of adopting the code form of "tool calling," underscoring the potential benefits of integrating programming paradigms into LLMs to bolster their understanding and execution of complex tasks. As shown in Table 2, the *Distill-Retrieve-Read* framework brings performance gains for the evidence retrieval process. Incorporated with the ability to distill dialogue history, RagPULSE is capable of summarizing the retrieval query. Compared with models whose number of parameters is less than 7 billion, RagPULSE (7B) demonstrates a notable performance enhancement in the context of retrieval accuracy, achieving at least a 30% improvement in document retrieval and a 15% increase in attribute retrieval according to HR@1 metrics. This shows that some of the current open-sourced LLMs still struggle with distilling key information from the long context to search for relevant evidence. Regarding the models with more parameters, RagPULSE (7B) still maintains a substantial lead, as evidenced by a 5% improvement in HR@1. Surprisingly, RagPULSE (7B) surpasses all models in attribute retrieval and RagPULSE (20B) performs better than GPT-3.5 (65.67 vs. 63.67 in document retrieval ). These results indicate that using "tool calling" to distill context benefits the query generation. To further validate the "tool calling"

| Model Name | Param. Size | Retrieved Doc. (%) | | | | Retrieved Attr. (%) | | | |
|---|---|---|---|---|---|---|---|---|---|
| | | HR@1 | HR@5 | HR@10 | HR@50 | HR@1 | HR@5 | HR@10 | HR@50 |
| History | - | 18.33 | 27.00 | 31.00 | 40.33 | 5.33 | 6.67 | 7.67 | 9.00 |
| Last Question | - | 28.33 | 35.00 | 37.67 | 40.00 | 12.33 | 15.67 | 16.33 | 17.67 |
| InternLM2 | 20B | 53.00 | 67.33 | 72.00 | 78.00 | 23.00 | 28.67 | 29.67 | 33.00 |
| PULSE | 7B | 53.00 | 62.67 | 66.00 | 70.33 | 18.00 | 21.00 | 22.00 | 23.33 |
| RagPULSE† | 7B | 58.67 | 69.67 | 75.67 | 78.67 | 19.67 | 22.67 | 23.67 | 25.00 |
| RagPULSE | 7B | 63.67 | 73.00 | 78.33 | 82.00 | 28.33 | 32.00 | 33.33 | 35.00 |
| QWen2 | 14B | 57.67 | 68.33 | 76.67 | 81.33 | 25.33 | 28.33 | 30.33 | 32.00 |
| RagQWen2 | 14B | 61.00 | 68.33 | 73.00 | 76.00 | 24.67 | 27.67 | 29.00 | 31.00 |
| PULSE | 20B | 56.33 | 66.33 | 69.67 | 74.00 | 22.00 | 26.33 | 26.67 | 28.00 |
| RagPULSE† | 20B | 60.33 | 70.67 | 75.00 | 81.00 | 29.33 | 34.00 | 34.67 | 38.67 |
| RagPULSE | 20B | 65.67 | 75.33 | 78.33 | 82.33 | 27.33 | 31.67 | 32.33 | 35.33 |

Table 3: Ablation of the *Distill-Retrieve-Read* framework. The "History" setting implements the retrieval process by using dialogue history as the query and the "Last Question" setting conducts searching via the last question. † represents the version where we use the same instruction for baseline models to prompt RagPULSE to generate the search query rather than using our proposed "tool calling" mechanism.

mechanism for summarizing the context, we also compare our RagPULSE with GPT-4, which is one of the most powerful LLMs. We can observe that RagPULSE achieves comparable results in generating search keywords with GPT-4 and performs better in precise retrieval (i.e., 65.67 vs. 62.33 in document retrieval and 28.33 vs. 26.67 in attribute retrieval).

Depending on the remarkable capabilities of PULSE in the medical field, RagPULSE achieves a higher score than other open-sourced models. To ablate the effect introduced by the relevant evidence, we directly use PULSE to respond to medical inquiries. Attributable to the specialized proficiency of PULSE in medical contexts, PULSE attains higher performance than other publicly available models. However, without utilizing retrieved evidence, the performance is not optimal. PULSE, referring to ground truth evidence of medicines (denoted as PULSE*), distinguishes itself from other models in the domain of medication consultation responses. This result highlights the challenge posed by medication consultation, which requires a vast amount of knowledge of medicine for practical application. We can see that RagPULSE outperforms all competing models and products in terms of responding to medication consultation, even with the retrieved evidence. This further validates the capability of the *Distill-Retrieve-Read* framework in generating accurate search queries for evidence retrieval in complex medical domains, reinforcing its value in boosting the performance of RAG-based LLMs in medication consultation scenarios.

## 4.3 ABLATION STUDIES

To fully investigate the contribution of our proposed *Distill-Retrieve-Read* framework, we conduct a quantitative analysis and report performances on MedicineQA when toggling the distillation part. The first two rows of Table 3 underscore the importance of distilling key information from dialogue history, which otherwise includes extraneous details detrimental to effective evidence retrieval. In addition, relying solely on the most recent query for information search proves inadequate due to the critical context embedded within the dialogue. Notably, RagPULSE (7B) exhibits more pronounced improvements, which outperforms PULSE (7B) with a notable 10% improvement.

Furthermore, as in the previous experiments, we also prompt our models to summarize the keywords without calling the tool. Compared with the PULSE without fine-tuning, RagPULSE† are observed to have significant performance gains in the two retrieval results. To empirically assess the effectiveness of our synthetic dataset, we conducted experiments with InternLM2 (20B) (Cai et al., 2024), which serves as the base model for PULSE (20B). We aimed to minimize interference from medical data. The results, as illustrated in the table, reveal that InternLM2 achieves outcomes comparable to PULSE. From the table, we can observe that InternLM2 achieves results comparable to PULSE. This indicates that merely fine-tuning medical domain data does not significantly enhance performance. However, RagPULSE demonstrated a significant improvement when utilizing our tool-calling dataset. The results validate the effectiveness of our proposed synthetic dataset for

| Model | RagPULSE 20B | RagPULSE 7B | GPT-3.5 | Baichuan2 | QWen2 | ChatGLM3 | MING | BenTsao | BianQue2 | DoctorGLM |
|---|---|---|---|---|---|---|---|---|---|---|
| Score | 60 | 46 | 50 | 34 | 32 | 30 | 28 | 0 | 0 | 0 |

Table 4: Human evaluation results to verify the effectiveness of our RagPULSE.

summarizing the history and confirm that fine-tuning models on our synthetic dataset can endow models with distillation abilities.

## 4.4 HUMAN EVALUATION

To further validate the performance of generating responses in the medical context, we conducted a human evaluation to annotate a subset of the generated answers. We recruited an additional five board-certified physicians to participate in the evaluation process. GPT-3.5 served as the baseline for comparison. For each question, the physicians were required to compare answers generated by other models with the baseline answer provided by GPT-3.5, assessing which answer was superior. For instance, for a given question (e.g., Question A), physicians needed to determine whether PULSE (20B) delivered a better response than GPT-3.5. Additionally, they were required to provide reasons supporting their judgments to enhance the validity of the evaluation. To ensure a fair comparison, we anonymized the names of the models and shuffled the order in which they were presented. The results from the five physicians were then aggregated to determine the final outcomes (The score of GPT-3.5 is set as 50). as can be seen in Table 4. The results indicate a high correlation between the Elo ratings and the human evaluations, suggesting the reliability of using Elo ratings for assessment.

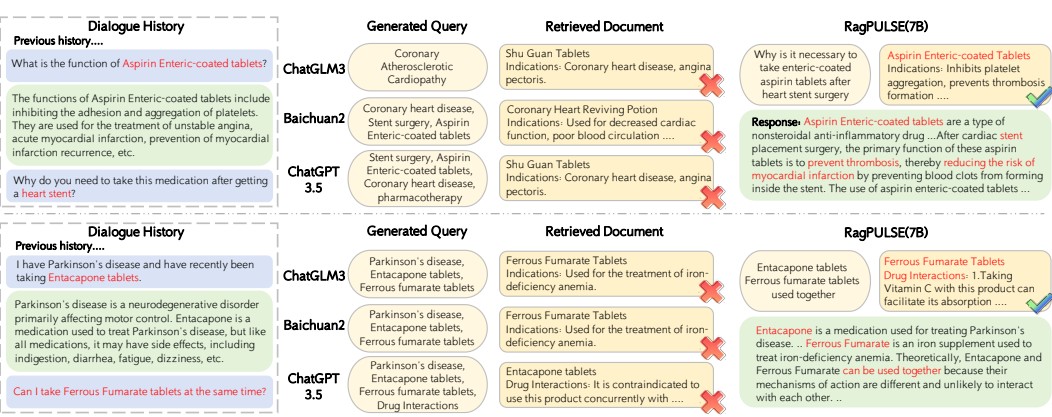

Figure 4: Case studies of LLMs' retrieval process and generated responses. LLMs first summarize the dialogue history and then generate search queries. The responses are formulated via the retrieved document. Key information is marked by red text.

## 4.5 CASE STUDY

To intuitively show how the *Distill-Retrieve-Read* framework makes a difference in the evidence retrieval process, we present examples (i.e., ChatGLM3, Baichuan2, GPT-3.5, and RagPULSE-7B) in Figure 4 to compare the generated searching queries and the retrieved evidence. As can be seen in the upper part, in scenarios involving lengthy history, extraneous information often leads to the generation of redundant and ineffective search queries. It is evident that, despite LLMs' ability to generate queries encapsulating all necessary information, the complexity of such queries frequently results in retrieval failures. In the lower part, although the query contains the corresponding medicine, the LLMs fail to understand the question, resulting in the omission of crucial keywords. Additionally, we can observe that GPT-3.5 still fails despite generating the correct keywords since the query does not contain key information about the question. These examples clearly indicate the state of current LLMs in the medication scenarios. With supplemented knowledge, RagPULSE shows hopeful performance in generating responses for medication consultation.

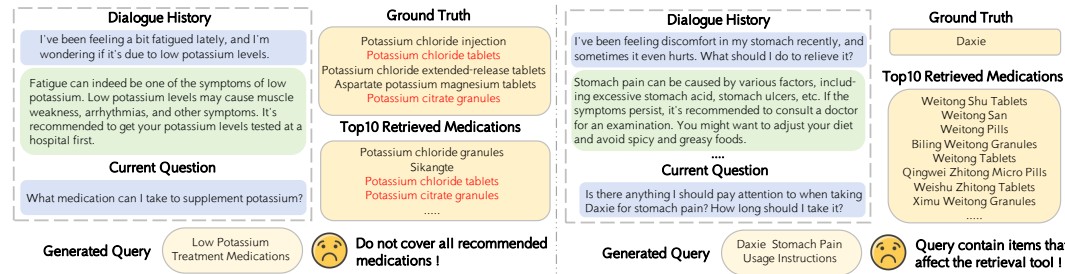

Figure 5: Failure cases of retrieving accurate medications. The failure modes into two main points: Do not cover all recommended medications (left), and Search queries containing items that affect the retrieval tool (right). Successfully retrieved medications are marked by red text.

## 5 ERROR ANALYSIS

To provide a comprehensive understanding of the distillation process, we present examples of retrieval failures in Figure 5. As shown in the left part, although the generated search query helps retrieve some correct evidence, the 10 retrieved pieces do not cover the ground truth, resulting in a retrieval failure. This phenomenon highlights the gap between domain-specific LLMs and clinical experts. More effort is needed to bridge this gap and bring these models closer to real-world applications. From the right part, we can observe that retrieval still fails even when the search query contains correct keywords. This failure can be attributed to certain keywords in the search process causing interference. When using the search query ['Daxie', 'Usage Instructions'], we can successfully retrieve the relevant evidence. However, physicians find the search query generated by RagPULSE to be more comprehensive, enabling a more precise search. Therefore, there is an urgent need to enhance the retrieval tool (an embedding model along with the authoritative database) to handle fine-grained medical terms effectively.

## 6 CONCLUSION

In this paper, we introduce MedicineQA, a new benchmark derived from real-world medication consultations, which aims to evaluate the capabilities of LLMs towards knowledge-intensive tasks in the medical domain. MedicineQA comprises 300 high-quality, expert-annotated multi-round dialogues spanning 10 key aspects of medication consultation scenarios. Along with the reference evidence, this pioneering work delves into exploring the evaluation of the retrieval process, illuminating a way of assessing the quality of search queries for retrieval-augmented generation (RAG). Our further study shows that the LLM with vanilla RAG is not enough for the medication consultation. To address this, we propose RagPULSE with a novel framework, *Distill-Retrieve-Read*, which revolutionizes the conventional *Retrieve-then-Read* through the innovative use of the "tool calling" mechanism. RagPULSE summarizes the intricate dialogue history and medication inquiry into the search query, mimicking human typing keywords for search engines. Extensive experiments demonstrate that our model gains superior performance compared to existing models in two evidence retrieval processes. Furthermore, integrated with an entity-oriented medicine database, our RagPULSE presents impressive results in responding to inquiries in medication consultation. We hope our work can motivate further innovation in applying LLMs in the medical domain.

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

# A APPENDIX

## A.1 DETAILS OF DATASET CREATION

Our dataset construction was conducted by a panel of 5 board-certified physicians, including a senior advisor overseeing the process. The other four physicians constructed the benchmark based on the selected records. They followed a set of detailed guidelines to ensure medical accuracy and relevance:

- Each question is crafted with contextual background aligned with medical knowledge and logic.
- Each question is designed to be realistic within the medical scenario, accurately corresponding to the preceding dialogue context.
- No identifiable information, except for age and gender, is disclosed in any question.

We also developed a comprehensive set of instructions focused on respecting patient privacy and maintaining the integrity of the medical information during the annotation process:

- Annotators received training on recognizing and handling sensitive information.
- They were instructed to remove any remnants of personal data they might encounter despite the initial automated de-identification process.
- Training also emphasized maintaining the contextual integrity of the medical advice while ensuring anonymity.
- Annotators received training on recognizing and handling sensitive information.

## A.2 DETAILS OF EVIDENCE RETRIEVAL

1. **Keyword Summary with GPT-4:** We use GPT-4 to summarize each question and related dialogue history into a set of key terms that encapsulate the core information and medical context of the inquiry.

2. **Searching the Medicine Database:** Utilizing the keywords generated by GPT-4, we query our extensive medicine database to collate a list of the 100 most relevant medications related to each query, ensuring the correct medication is within the top 100 identified.

3. **Constructing Answers:** Four board-certified physicians independently reviews the list of 100 medications and constructs a potential answer based on their medical expertise and the relevance of the medication to the query.

4. **Voting on the Best Answer:** Once all proposed answers are submitted, a voting process ensues where the answer deemed most accurate and appropriate for the question is selected.

5. **Resolving Ties with the Senior Advisor:** In cases where there is a tie in the voting, the senior advisor intervenes to review the question and tied answers for any potential issues. If a problem is identified with the question itself, it is sent back for reconstruction. If the question is deemed appropriate, the senior advisor then evaluates the tied answers based on medical accuracy and relevance, scoring each to determine the highest-quality response. Alternatively, all five board-certified physicians may reconvene to discuss and agree upon the best answer, ensuring that the final selection is reached through consensus and expert validation.

6. **Finalization of the QA Pair:** The answer that emerges from this process—either through direct voting, senior advisor evaluation, or a full panel discussion—is then paired with the original question to form a finalized QA pair in the MedicineQA dataset.

## A.3 DETAILS OF ELO

The Elo rating system, devised by Arpad Elo, is a methodical framework used to calculate the relative skill levels of players in competitor-versus-competitor games. Initially conceived for chess, the Elo system has found widespread application across various sports and games to gauge individual

or team performance. The fundamental principle of the Elo system is to assign a numerical rating to each player, which adjusts based on match outcomes against other rated players. The adjustment in ratings is predicated on the difference between the actual and expected match outcomes, allowing for a dynamic representation of the skill level over time.

The core of the Elo rating system is encapsulated by the formula used to update player ratings post-match. The expected score for a player, $E_A$, against an opponent, is calculated as:

$$E_A = \frac{1}{1 + 10^{(R_B - R_A)/400}}$$

where $R_A$ and $R_B$ are the current ratings of the player and the opponent, respectively. Following the completion of a match, the actual score $(S_A)$ − 1 for a win, 0.5 for a draw, and 0 for a loss -is compared against the expected score to update the player's rating:

$$R'_A = R_A + K\left(S_A - E_A\right)$$

In this formula, $R'_A$ represents the new rating of the player, and $K$ is a factor that determines the maximum possible adjustment per game. This factor can vary depending on the level of competition and the governing body's regulations, allowing for flexibility in the sensitivity of rating adjustments to match outcomes. The Elo system's adaptability and simplicity have contributed to its enduring popularity and applicability across different competitive disciplines.

