# OpenReview forum: "Tool Calling: Enhancing Medication Consultation via Retrieval-Augmented Large Language Models"
_ICLR.cc/2025/Conference — ICLR 2025 Conference Withdrawn Submission_

### Official Review · Reviewer_5Hq1 · 2024-11-01

**Soundness:** 4
**Presentation:** 4
**Contribution:** 3
**Rating:** 6
**Confidence:** 4

**Summary:**

The paper has introduced a benchmark called 'MedicineQA' to evaluate llms in medical consultation. Their suggested benchmark is focusing on rag for multi-turn dialogues in medication consultation. The authors also propose RagPULSE framework to deal with limitations of standard RAG frameworks. This framework utilizes a tool-calling mechanism to enhance retrieval accuracy by summarizing dialogue history into search queries and retrieving evidence from a medicine database.

**Strengths:**

This paper shows an approach to enhancing LLMs in the medical domain by reformulating rag for multi-round medication consultations. Also, using a tool-calling mechanism for search query formulation is a novel method of tailoring retrieval processes in dialogue-based tasks in healthcare. The paper shows good quality in methodology. Also, the authors did around 600 experiments to validate the effectiveness of ragpulse and provided comparisons with existing open-source and commercial models. The paper has clear objectives, methods, and findings sections written. The study addresses a research gap in applying LLMs to the medical field and introduced MedicineQA, ragpulse. Their work shows good retrieval and response generation capabilities.

**Weaknesses:**

I feel that the number of multi-round dialogues (300) are very less and that makes it difficult in terms of generalizing the results. So, maybe increase the dataset size by expanding the set of topics like for various specific conditions like breast cancer, diabetes, chronic pain.

Another thing is that the study should discuss retrieval errors for the cases where the model fails to retrieve accurate responses. The authors can try to include 'error analysis' section (maybe) that show various retrieval failures types or examples. This analysis will certainly go a long way in understanding if any adjustments should be made in distillation process.

**Questions:**

1. How ragpulse deals with retrieval errors (if any)? Also, did you see any failure modes commonly found and can you discuss them, please?

2. The benchmark is available in English and Chinese as per the paper. but how well do we know if MedicineQA can be adapted to other languages (maybe low-resource languages)?

---

> ### Author Response · Authors · 2024-11-18
>
> Thank you for your constructive comments. We will address your concerns in the following parts.
>
> # Limited data size
>
> Thank you for your valuable feedback. The current version of MedicineQA focuses on commonly used medicines to address a broad range of prevalent needs. Although current MedicineQA contains only 300 samples, it covers most of the medicines commonly used in real-world scenarios, spanning 10 aspects of medicine application. We appreciate your suggestion to increase the dataset size by expanding the set of topics for various specific conditions, which is crucial for improving the scope of our benchmark. We sincerely plan to include more specialized medications in the future. However, the construction process is time-consuming.
>
> The development of MedicineQA involved significant input from medical experts, with each step being guided by real-world data and supervised by physicians to ensure the benchmark’s quality. Additionally, expanding the dataset requires incorporating more medicines from the original, large-scale Medicine Database into our knowledge base. This process involves rigorous review and verification. We are currently in the process of reviewing additional medicines and expanding the dataset. We regret any delays in providing a larger benchmark and sincerely appreciate your understanding and patience.

---

> > ### Author Response · Authors · 2024-11-18
> >
> > # Error analysis
> >
> > We appreciate your suggestions. We have analyzed the failure cases of RagPULSE and summarized the failure modes into two main points: 1. Failures in the aspect of Medication Recommendation, and 2. Search queries containing items that affect the retrieval tool. To provide a comprehensive understanding of the distillation process, we present examples of retrieval failures in the following tables.
> >
> > | Dialogue History                                             | Current Question                                             | Generated Search Query                 | Retrieved Evidence (Top 10)                                  | Ground Truth                                                 |
> > | ------------------------------------------------------------ | ------------------------------------------------------------ | -------------------------------------- | ------------------------------------------------------------ | ------------------------------------------------------------ |
> > | **User**: I've been feeling a bit fatigued lately, and I'm wondering if it's due to low potassium levels. **Medical Assistant**: Fatigue can indeed be one of the symptoms of low potassium. Low potassium levels may cause muscle weakness, arrhythmias, and other symptoms. It's recommended to get your potassium levels tested at a hospital first. | What medication can I take to supplement potassium?          | Low Potassium Treatment Medications    | ['Potassium chloride granules', 'Sikangte', '**Potassium chloride tablets**', '**Potassium citrate granules**', 'Potassium citrate oral solution', 'Pingjiabao', 'Shiertong', 'Potassium chloride extended-release tablets', 'Potassium iodide tablets', 'Rativiru potassium tablets'] | ['Potassium chloride injection', '**Potassium chloride tablets**', 'Potassium chloride extended-release tablets', 'Aspartate potassium magnesium tablets', '**Potassium citrate granules**'] |
> > | **User**: I've been feeling discomfort in my stomach recently, and sometimes it even hurts. What should I do to relieve it?**Medical Assistant**: Stomach pain can be caused by various factors, including excessive stomach acid, stomach ulcers, etc. If the symptoms persist, it's recommended to consult a doctor for an examination. You might want to adjust your diet and avoid spicy and greasy foods.**User**: The stomach pain is intermittent. I ate some light food and felt a little better. **Medical Assistant**: A light diet can help relieve the burden on the stomach. If the pain doesn't improve or worsens, you may consider using medications that protect the stomach lining. Daxie can help relieve stomach pain caused by excessive stomach acid.**User**: Does Daxie have any side effects? **Medical Assistant**: Daxie is a common antacid medication, and it usually has minimal side effects. However, long-term or excessive use may cause constipation, bloating, and other issues. If you experience discomfort, it's important to stop taking it and consult a doctor. | Is there anything I should pay attention to when taking Daxie for stomach pain? How long should I take it? | Daxie  Stomach Pain Usage Instructions | ['Weitong Shu Tablets', 'Weitong San', 'Weitong Pills', 'Biling Weitong Granules', 'Weitong Tablets', 'Qingwei Zhitong Micro Pills', 'Weishu Zhitong Tablets', 'Ximu Weitong Granules', 'Xiangsha Weitong San', 'Jianwei Zhitong Tablets'] | ['Daxie']                                                    |
> >
> > In MedicineQA, there are multiple ground-truth evidence entries for the aspect of Medication Recommendation. We adopt a strict evaluation metric: Assume the number of retrieved evidence $E$ is $x$ and the number of ground truth $G$ is $y$. If $x \leq y$, retrieval is considered successful only when $E \subseteq G$. If $y \leq x$, retrieval is considered successful only when $G \subseteq E$. In the first row, although the generated search query helps retrieve some correct evidence, the 10 retrieved pieces do not cover the ground truth, resulting in a retrieval failure. This phenomenon highlights the gap between domain-specific LLMs and clinical experts. More effort is needed to bridge this gap and bring these models closer to real-world applications.
> >
> > In the second row, we observe that retrieval still fails even when the search query contains correct keywords. This failure can be attributed to certain keywords in the search process causing interference. When using the search query ['Daxie', 'Usage Instructions'], we can successfully retrieve the relevant evidence. However, physicians find the search query generated by RagPULSE to be more comprehensive, enabling a more precise search. Therefore, there is an urgent need to enhance the retrieval tool (an embedding model along with the authoritative database) to handle fine-grained medical terms effectively.
> >
> > Thank you again for your advice on adding error analysis. We will include this section in the revised paper.

---

> > > ### Author Response · Authors · 2024-11-18
> > >
> > > # Extend to other languages
> > >
> > > Thank you for your question. We are very glad to adapt MedicineQA to other languages, especially for low-resource languages, as our primary goal is to cover commonly used medicines to meet a wide range of prevalent needs and support regions with relatively scarce medical resources. We are willing to translate MedicineQA into other languages. Additionally, we are willing to provide proofreading and translation services for a subset the database  to ensure comprehensive support. We appreciate your understanding and patience.

---

> > > ### Comment · Reviewer_5Hq1 · 2024-11-26
> > >
> > > Thank you for answering my questions. I will increase my score by 1 point.

---

> > > > ### Author Response · Authors · 2024-11-26
> > > > **Thanks for your encouraging feedback.**
> > > >
> > > > We are deeply grateful for your consideration of our responses and for raising your score. Your feedback has been invaluable in refining our work.
> > > >
> > > > If you have any additional suggestions or concerns, we would be more than happy to address them to further enhance the quality of our manuscript.
> > > >
> > > > Thank you once again for your support and encouragement.

---

> ### Author Response · Authors · 2024-12-04
>
> Dear Reviewer 5Hq1,
>
> We would like to sincerely thank you for your time and effort in engaging during the discussion phase. We greatly appreciate the constructive feedback and thoughtful comments you provided, which have been invaluable in improving the quality of our work.
>
> Best regards,
>
> Authors

---

### Official Review · Reviewer_gvnx · 2024-11-05

**Soundness:** 2
**Presentation:** 3
**Contribution:** 2
**Rating:** 5
**Confidence:** 4

**Summary:**

This paper introduces a benchmark, database and method for describing medications in dialogues with LLMs. The benchmark is collected automatically and reviewed by medical experts and the database is curated to support retrieval of medication information. The paper evaluates the proposed approach of converting conversations with their histories into a set of keywords to query the medical database and the results indicate favourable outcomes, based on GPT-4- and human-based evaluations of the generations.

**Strengths:**

The collected resources seem helpful and, even if collected automatically, are validated by domain experts. The proposed approach seems to improve over LLM-based baselines.

The experiments seem sound in what they report/investigate.

The overall writing quality is good.

**Weaknesses:**

My main concerns is the lack of meaningful comparison and under-specification of some of the details.

- While the paper suggests that the proposed distill-then-retrieve framework is better than RAG methods, I did not find meaningful comparisons with either general RAG frameworks as e.g. discussed in lines 117-126, or with domain-specific RAG approaches, such as e.g. BioReader (https://aclanthology.org/2022.emnlp-main.390.pdf) RAFT (https://arxiv.org/abs/2403.10131).

- The experimental setting is somewhat under-specified, see my questions to the authors. In summary, I don't understand the exact settings for the comparisons in Tables 2 and 3.

- There is very little detail on the construction of the medical database, which seems to be the key contribution enabling the retrieve-then-distill framework. Particularly, data sources, data quality measures etc are missing.

I gravitate towards rejection but if my queries are answered with sufficient detail, I am open to re-evaluating my decision.

**Questions:**

1) What is the use of the question annotations in MedicineQA? If I understand correctly, it is the explicit reformulation of the questions from the conversation history (which might be ambiguous and contain superfluous information, as shown in "Dialogue History" in Figure 3). I understand that the synthetic dataset collected and described in lines 257-264 is used to convert questions into keywords, but during inference, what is actually used as input? Is it the dialogue history, of the extracted questions. If the latter, how are they extracted from dialogue history?

2) Could you please indicate how you measure the H@K for "baseline models". From reading lines 300-306, It is not clear to me what the baselines are actually doing. Are they generating the medication names from their parameters? Are they generating queries that are run against the database?

3) Furthermore, could you please elaborate on what you mean by "we immediately adopt the attached corresponding medicine information as the context to guide the generation [...]" in line 323. Is it the ground-truth medicine annotations from MedicineQA? If so, I find this result highly counter-intuitive and contradictive, as the paper states later, in lines 389 ff, that the improvements of RagPULSE are due to the proposed RAG framework rather than due to domain fine-tuning of the PULSE LLM. But since RagPULSE's retrieval is far from optimal (e.g. H@k rates in Tables 2 and 3), it would suggest that imperfect retrieval somehow leads to better generations. Similarly, could you please indicate what "instruction used for baseline models" in the description of Table 3 means: "We also prompt RagPULSE by the instruction used for baseline models, which are denoted as †".

---

> ### Author Response · Authors · 2024-11-18
>
> Thank you for your constructive comments. We will address your concerns in the following parts.
>
> # W1: Limited Comparison
>
> We apologize for any confusion caused by the unclear motivation. Our primary focus is to address the unique challenges of medication consultation scenarios. Medication consultation is a knowledge-intensive task that requires an authoritative medicine database to provide relevant evidence for generating correct and comprehensive responses without hallucinations. In real-world scenarios, dialogues are often ambiguous and verbose. For example, users may use layman's terms rather than medical terminology and provide more information than what is medically relevant. As demonstrated in Table 3 of our submitted paper, directly utilizing the user's query or the entire dialogue history yields poor results. Thus, effectively summarizing user inputs and eliminating extraneous information to retrieve appropriate evidence from the medicine database is crucial. To address this, we propose a novel approach for generating a simple yet robust search query, which we refer to as the "Distill" step in our *Distill-Retrieve-Read* framework. Specifically, we prompt the LLM to generate search keywords for several predefined search engines. When a user query comes, the LLM mimics how a human would use search engines, generating appropriate search keywords based on the current dialogue context. This approach differs from the current mainstream methods, which typically prompt LLMs to summarize the previous context into search keywords. Recent studies have demonstrated the impressive capabilities of LLMs in summarizing contexts into keywords. Following the previous premise, we conduct experiments by comparing our proposed "Distill" method with methods that prompt LLMs to summarize. As shown in Table 2 in our paper, our "Distill" method demonstrates superior performance over directly prompting LLMs to summarize.
>
> Thank you again for your valuable feedback.  Your recommended approaches have truly inspired us. Previous works in RAG have typically focused on improving retrieval accuracy through specially designed frameworks or extracting useful information from dummy documents. We are regretful that few have addressed the critical challenge of generating more accurate search keywords. To the best of our knowledge, MedicineQA, with our medicine database, is the first benchmark in the medical domain that evaluates the robustness of generated search keywords. Since the medicine database is constructed in Chinese, it would be time-consuming to translate this huge database into English while ensuring its correctness. It is unfair to compare models in English with a raw medicine database.  Thus,  we also compare our RagPULSE with GPT-4, which is one of the most powerful LLMs. As shown in the following table. We can observe that RagPULSE achieves comparable results in generating search keywords with GPT-4 and performs better in precise retrieval (i.e., 65.67 vs. 62.33 in document retrieval and 28.33 vs. 26.67 in attribute retrieval).
>
> | Model Name | Param.Size | Ins.follow rate | Doc.HR@1 | Doc.HR@5 | Doc.HR@10 | Attr.HR@1 | Attr.HR@5 | Attr.HR@10 |
> | :--------: | :--------: | :-------------: | :------: | :------: | :-------: | :-------: | :-------: | :--------: |
> |   GPT-4    |     -      |     100.00      |  62.33   |  76.33   |   82.00   |   26.67   |   32.33   |   34.00    |
> |  RagPULSE  |     7B     |     100.00      |  63.67   |  73.00   |   78.00   |   28.33   |   32.00   |   33.33    |
> |  RagPULSE  |    20B     |     100.00      |  65.67   |  75.33   |   78.33   |   27.33   |   31.67   |   32.33    |

---

> ### Author Response · Authors · 2024-11-18
>
> # W2: Details of Experiments
>
> We apologize for any confusion caused by the omission of certain details. We hope the following clarifications will provide a clearer understanding of our methodology.
>
> **Q1: Illustration of the workflow via the example in Figure 3:** Each sample in MedicineQA consists of three parts: (*Dialogue History* $H$, *Current Question* $Q_{T+1}$, *Relevant Evidence* {$K_c$, $K_f$}). During the construction process, physicians summarized selected records into a question ($Q_{T+1}$) about a specific medicine. The *Dialogue History* ($H$) is generated by GPT-4 and further revised by physicians. The *Relevant Evidence* ($K_c$, $K_f$) is annotated by experts, with more details available in the Appendix. We apologize again for any confusion caused by the ambiguous figure.
>
> Specifically, *Dialogue History* includes the previous user input, such as "I am overweight..." and the previous model response, "Note that the following information...". *Current Question* is, for example, "How many pills...". The *Prompt for Distillation* is the *System Prompt* for RagPULSE, designed to instruct the LLM that multiple search engines can be used to retrieve information. To search for relevant information from the Medicine Database, RagPULSE uses both *Dialogue History* and *Current Question* as inputs, summarizing them into a search query (e.g., "Orlistat Capsules: Dosage, Usage, and Adverse Reactions"). In practice, the output of RagPULSE is *search engine(Orlistat Capsules: Dosage, Usage, and Adverse Reactions)*, where the model mimics the human process of using search engines.
>
> **Q2: Illustration of settings for baseline models:** For the baseline models, we first prompt them to summarize the *Dialogue History* $H$ and *Current Question* $Q_{T+1}$ into a search query using the instruction (Based on the above conversation about medical inquiries and medication queries, please summarize the search keywords for the user's final question using the dialogue record. Retrieve relevant medication information and return it in JSON format as follows:\{"query'': $\dots$\}) . The generated search query is then used to query the database to retrieve evidence. Based on the retrieved evidence, we measure the HR@K for the baseline models according to the *Relevant Evidence*.
>
>
>
> **Q3: Elaboration of the results:** We are very grateful for your careful reading. As shown in Table 2, some baseline models exhibit poor performance in generating accurate search queries for retrieving evidence from the medicine database. To evaluate the quality of the generated responses based on the relevant evidence, we immediately adopt the corresponding ground-truth medicine information (i.e., the annotations from MedicineQA, as you mentioned) as the context to guide the response generation. We apologize again for any confusion in line 389.  In this section, we aimed to investigate whether fine-tuning on the synthetic dataset could improve the model's ability to summarize context. It is important to note that both PULSE and RagPULSE are domain-specific models. Apart from the synthetic dataset, all other fine-tuning data are the same as those used for PULSE.  RagPULSE† represents the version where we use the same instruction for baseline models to prompt RagPULSE to generate the search query rather than using our proposed "Distill" step. The comparisons between PULSE and RagPULSE† in Table 3 indicate that finetuning on the synthetic dataset contributes to enhancing the model's capability to summarize context effectively.
>
> In Table 2, RagPULSE leverages the retrieved evidence to generate the answer. While the retrieval process in RagPULSE is not optimal, it still achieves notable performance in combining the evidence to generate the final response. This can be attributed to PULSE’s specialized proficiency in the medical domain. To address your concerns, we present additional results showing PULSE without evidence and PULSE with the ground-truth medicine annotations from MedicineQA in the following table (More results can be seen in Table 1 of the revised paper).  PULSE attains higher performance than other publicly available models. However, without utilizing retrieved evidence, the performance is not optimal. PULSE*, referring to ground truth evidence of medicine, distinguishes itself from other models in the domain of medication consultation responses.
>
> | Model      | PULSE*(20B) with annotations from MedicineQA | RagPULSE(20B) | RagPULSE(7B) | PULSE(20B) |
> | ---------- | -------------------------------------------- | ------------- | ------------ | ---------- |
> | Elo Rating | 1094                                         | 1074          | 1060         | 1041       |

---

> ### Author Response · Authors · 2024-11-18
>
> # W3: Details of the Medicine Database
>
>  We purchased a comprehensive Chinese medicine database from an authoritative institution as a foundation for our RAG model. It includes detailed descriptions of hundreds of thousands of medications, encompassing both traditional Chinese and Western medicines. Each medication document has been subjected to a rigorous triple review and verification process to ensure compliance with established medical standards. We are currently in the process of translating and validating subsets of the database. We regret any delays in making this resource available as an open-source tool and appreciate your understanding.

---

> > ### Author Response · Authors · 2024-11-25
> > **Sincerely looking forward to more discussion with you**
> >
> > Dear Reviewer gvnx,
> >
> > Thank you for your valuable time and insightful comments on our submission. We have carefully addressed your concerns by providing detailed responses and corresponding results, which we believe adequately cover the issues you raised.
> >
> > We would greatly appreciate the opportunity to further discuss whether your concerns have been fully addressed. If there are any remaining points or additional clarifications needed, we are more than willing to provide further information.
> >
> > Thank you once again for your time and support.
> >
> > Best regards,
> > Authors

---

> ### Author Response · Authors · 2024-11-27
>
> Dear Reviewer gvnx,
>
> We sincerely appreciate your time and effort in reviewing our work. We thank you for your deep comprehension and suggestions, which have strengthened our manuscript. We would appreciate the opportunity to continue dialogue with you to fully meet your expectations. If you still find some aspects of our work unclear or have other additional concerns, we are always prepared to discuss or provide more information. Our commitment is to address any concerns you may have and enhance the quality of our paper.
>
> Best regards,
> Authors

---

> > ### Author Response · Authors · 2024-12-01
> >
> > Dear Reviewer gvnx,
> >
> > We deeply appreciate the time and effort you have invested in reviewing our work and providing insightful questions and suggestions. In our response, we have clarified the points regarding meaningful comparisons, experimental settings, the medicine database, and the key questions you raised about the use of annotations, the H@K metric, and RagPULSE’s performance details. We are currently in the process of translating and validating subsets of the database. We hope this database and our benchmark would help further research. We regret any delays in making this resource available as an open-source tool and appreciate your understanding.
> >
> > We have taken great care to ensure our clarifications are thorough and directly address your concerns and updated the revised version of our manuscript. If there are any additional points requiring elaboration or areas you would like us to further clarify, please do not hesitate to let us know. We would be delighted to continue communicating with you.
> >
> > Best regards,
> >
> > Authors

---

> ### Comment · Reviewer_gvnx · 2024-12-03
>
> Dear authors,
>
> thank you for your answers. I still think the paper is confusing to read and would benefit from a rewrite - I also don't think your answers did a good job addressing my questions. Therefore, I would like to maintain my rating. However, if the other two reviewers would like to champion this paper, I am not opposed to acceptance either.
>
> On an off note, and this is supposed to be a well-meant advice and has no influence on my assessment of your submission: When addressing reviewer questions, it's best to be precise and to the point, to help the reviewer find the answers they're looking for. A wall of text is not very helpful in this case. With ChatGPT, everyone can write a long-winded answer. The tricky part is to present the necessary information in a concise manner. Sorry to be blunt, but when I'm looking for answers to my questions, I really don't care whether my "recommended approaches have truly inspired" you.

---

> > ### Author Response · Authors · 2024-12-03
> >
> > Dear Reviewer gvnx,
> >
> > Thank you for your feedback. We have revisited and updated our response document to ensure that each of your concerns is addressed more precisely and concisely. Thank you again for your time and insights. We would be delighted to continue communicating with you.
> >
> > Best regards,
> >
> > Authors

---

> > > ### Comment · Reviewer_gvnx · 2024-12-03
> > >
> > > Okay, you've added headers and boldfaced paragraph titles, but my main question still remains unresolved:
> > >
> > > how do you reconcile that generating based on imperfect retrieval with a domain specific model (RagPULSE) leads to better performance than generating with a general model based on retrieving with ground truth?
> > >
> > > What I take away from your experimental results, is that the domain-specific fine-tuning (which is not your contribution) seems to have more impact than the proposed RAG mechanism (which is your contribution). So why would I care for RAG when I can just use an existing domain-finetuned model?

---

> > > > ### Author Response · Authors · 2024-12-03
> > > >
> > > > Thank you once again for your time. We hope the following illustration would address your concerns:
> > > >
> > > > 1. **Motivation for Medication Consultation Task**
> > > >    Medication consultation is a knowledge-intensive task that requires an authoritative medicine database to provide relevant evidence for generating correct and comprehensive responses without hallucinations.  In real-world scenarios, **existing domain-finetuned models lack the capacity to encompass the extensive knowledge of medicines needed to address the diverse questions posed by users.** Towards knowledge-intensive tasks, existing methods incorporated external knowledge with LLMs by retrieval augmentation. Therefore, retrieving appropriate evidence from a medical database is critical for LLMs to generate high-quality responses in this scenario. This underscores the importance of a robust retrieval-augmented generation (RAG) approach in enhancing the model's performance for medication consultation.
> > > >
> > > > 2. **Clarification Regarding Table 2 (Q3)**
> > > >    To address your concerns, we have presented additional results showing PULSE without evidence and PULSE with the ground-truth medicine annotations from MedicineQA in Table 2 of the revised paper. There may have been some misunderstandings about the Elo rating system used in Table 2 of the revised paper. When a new model is added, the ratings for all models are recalculated. The revised results in Table 2 are as follows:
> > > >
> > > >    | Model Name     | Elo Rating | Elo Rank |
> > > >    | -------------- | ---------- | -------- |
> > > >    | PULSE* (20B)   | 1094       | 1        |
> > > >    | RagPULSE (20B) | 1074       | 2        |
> > > >    | GPT-3.5        | 1068       | 3        |
> > > >    | RagPULSE (7B)  | 1060       | 4        |
> > > >    | PULSE (20B)    | 1041       | 5        |
> > > >    | Baichuan2      | 1037       | 6        |
> > > >    | MING           | 1002       | 7        |
> > > >    | QWen2          | 998        | 8        |
> > > >    | ChatGLM3       | 979        | 9        |
> > > >    | DoctorGLM      | 896        | 11       |
> > > >    | BenTsao        | 889        | 11       |
> > > >    | BianQue2       | 862        | 12       |
> > > >
> > > >    The reason RagPULSE demonstrates strong performance can be attributed to the specialized domain-specific fine-tuning of PULSE. **However, as shown in the Table, without utilizing retrieved evidence, the performance of PULSE is not optimal.**  Moreover, PULSE, referring to ground truth evidence of medicines (denoted as PULSE$^*$), distinguishes itself from other models in the domain of medication consultation responses. This result highlights the challenge posed by medication consultation, which requires a vast amount of knowledge of medicine for practical application.
> > > >
> > > >
> > > > 3. **Rationale for Choosing PULSE as Baseline**
> > > >    In this work, we selected PULSE (a domain-specific fine-tuning model) as our baseline because **domain-specific fine-tuning models are better equipped to understand the context with medical knowledge, which facilitates the subsequent RAG process.** This capability is crucial for addressing the nuanced requirements of medication consultation tasks.
> > > >
> > > > Thank you again for your valuable feedback. We would appreciate the opportunity to continue dialogue with you to fully meet your expectations. If you still remain some aspects of our work unclear or have other additional concerns, we are always prepared to discuss or provide more information to you.

---

> > > > > ### Author Response · Authors · 2024-12-03
> > > > >
> > > > > We hope the following case will further demonstrate the limitations of domain-finetuned models in medication consultation and address your concerns.
> > > > >
> > > > > For instance, when a user inquires about the dosage of *YiKeXin* Vitamin AD Drops, the responses differ significantly between LLMs without Retrieval-Augmented Generation (RAG) and those with RAG integration:
> > > > >
> > > > > ------
> > > > >
> > > > > ### LLMs without RAG:
> > > > >
> > > > > ***YiKeXin* Vitamin AD Drops** (suitable for children) dosage is usually determined based on age. However, specific dosages should be adjusted according to recommendations from a doctor or pharmacist. Below is a general guideline:
> > > > >
> > > > > #### Neonates and Infants (0-3 years old)
> > > > >
> > > > > - **0-1 year**: One capsule daily (**400 IU of Vitamin A and 200 IU of Vitamin D**).
> > > > > - **1-3 years**: One capsule daily (**700 IU of Vitamin A and 400 IU of Vitamin D**).
> > > > >
> > > > > #### Additional Notes
> > > > >
> > > > > 1. **Timing of Administration**: Typically recommended with or after meals to aid absorption.
> > > > > 2. **Avoid Overdosage**: Both Vitamin A and D are fat-soluble vitamins; overdosage can cause toxicity. Avoid taking multiple supplements containing Vitamin AD simultaneously.
> > > > > 3. **Special Cases**: Dosage may require adjustment under medical supervision for preterm infants, low-birth-weight infants, or children with vitamin deficiencies.
> > > > >
> > > > > However, the following part contains **hallucinations, illustrating incorrect dosage and internal contradictions regarding different specifications for the same medication**:
> > > > >
> > > > > - **0-1 year**: One capsule daily (**400 IU of Vitamin A and 200 IU of Vitamin D**).
> > > > > - **1-3 years**: One capsule daily (**700 IU of Vitamin A and 400 IU of Vitamin D**).
> > > > >
> > > > > ### LLMs with RAG:
> > > > >
> > > > > The dosage of *YiKeXin* Vitamin AD Drops varies based on the child’s age:
> > > > >
> > > > > - **Infants under 1 year old**: Once daily, one capsule per dose.
> > > > > - **Children over 1 year old**: Once daily, one capsule per dose.
> > > > >
> > > > > #### Instructions for Use:
> > > > >
> > > > > Open the soft capsule nozzle and directly drip the contents into the infant’s mouth. It is recommended to immerse the nozzle in hot water for 30 seconds to soften the rubber, making it easier to open.
> > > > >
> > > > > Please strictly follow the recommended dosage and do not exceed it. If other medications are being used or if there are specific health conditions, consult a physician or pharmacist before use.
> > > > >
> > > > > ------
> > > > >
> > > > > This case highlights the hallucinations present in domain-finetuned models without RAG, which can result in inconsistent and potentially harmful recommendations. By contrast, models with RAG integration provide more accurate and practical advice by leveraging reliable sources. We hope this clarifies the differences and addresses your concerns effectively.

---

> > > > > ### Comment · Reviewer_gvnx · 2024-12-03
> > > > > **OK**
> > > > >
> > > > > Ok, I think this explanation makes more sense - but really the useful comparisons are those that include PULSE. I feel like the inclusion of the non-domain-specific LLMs is a bit of a red herring here, even with ground-truth retrieved evidence they are not perfoming better than a domain-fine-tuned model (i.e. PULSE). I take your point that RagPULSE improves upon PULSE, but it still seems to me that overall, the main advantage comes from domain adaptation, not from the proposed RAG framework.

---

> ### Author Response · Authors · 2024-12-03
>
> Thank you once again for your time and valuable feedback. Specifically, with the exception of Baichuan2, Qwen2, and the GPT series, the other models are domain-specific for the medical field. **The suboptimal performance of these models, even with ground-truth retrieved evidence, further highlights the challenges present in medication consultation scenarios.**
>
> In fact, domain-specific fine-tuning can indeed enhance an LLM's ability to understand medical contexts and generate responses. However, it contributes little to the summarization and keyword generation necessary for effective evidence retrieval. **As shown in the table below, InternLM, the base model of PULSE, actually outperforms PULSE in generating search queries for retrieval.** When further augmented with the *Distill* approach, RagPULSE shows a significant improvement. The results validate the effectiveness of our proposed framework. The primary reason for selecting PULSE is that it is open-source and available for fine-tuning with the existing dataset.
>
> | Model Name   | Param. Size | HR@1 (Doc) | HR@5 (Doc) | HR@10 (Doc) | HR@50 (Doc) | HR@1 (Attr.) | HR@5 (Attr.) | HR@10 (Attr.) | HR@50 (Attr.) |
> | ------------ | ----------- | ---------- | ---------- | ----------- | ----------- | ------------ | ------------ | ------------- | ------------- |
> | InternLM 20B | 20B         | 53.00      | 67.33      | 72.00       | 78.00       | 23.00        | 28.67        | 29.67         | 33.00         |
> | PULSE 20B    | 20B         | 56.33      | 66.33      | 69.67       | 74.00       | 22.00        | 26.33        | 26.67         | 28.00         |
> | RagPULSE 20B | 20B         | 65.67      | 75.33      | 78.33       | 82.33       | 27.33        | 31.67        | 32.33         | 35.33         |
>
> We hope our response could address your concerns.

---

> > ### Author Response · Authors · 2024-12-04
> >
> > Dear Reviewer gvnx,
> >
> > We would like to sincerely thank you for your time and effort in engaging during the discussion phase. We greatly appreciate the constructive feedback and thoughtful comments you provided. We hope that our efforts can address your concerns.
> >
> > Best regards,
> >
> > Authors

---

### Official Review · Reviewer_rJMn · 2024-11-05

**Soundness:** 2
**Presentation:** 3
**Contribution:** 2
**Rating:** 6
**Confidence:** 3

**Summary:**

This paper built a multiple-round dialogue benchmark MedicineQA and proposed a distill-retrieve-read framework for RAG improved medication consultation capability of LLM. The dataset size is limited, and the multiple-round dialogues were generated by GPT-4 but not from real-world dialogue; this may affect the value of the application for the proposed framework. More details about the dataset should be introduced, such as the number of dialogue rounds.  It seems the dialogue distilling tool was trained on the general domain synthetic dataset but not for the medical domain, and the details of the dialogue distill tool training are not clear (which base model? data size? training settings? list of data sources). The proposed method lacks novelty; it applies the dialogue distilling module in the RAG setting and tests on the small MedicineQA dataset. Experiments are only based on a single dataset.
Typo? The manuscript mentioned "Table 3.2," but this is not Table 3.2; I guess it should be Table 1.
Question: Will the medicine database be released to the public?

**Strengths:**

This paper proposed a new dataset for medication consultation.

**Weaknesses:**

The dataset size is limited, and the multiple-round dialogues were generated by GPT-4 but not from real-world dialogue; this may affect the value of the application for the proposed framework. More details about the dataset should be introduced, such as the number of dialogue rounds.  It seems the dialogue distilling tool was trained on the general domain synthetic dataset but not for the medical domain, and the details of the dialogue distill tool training are not clear (which base model? data size? training settings? list of data sources). The proposed method lacks novelty; it applies the dialogue distilling module in the RAG setting and tests on the small MedicineQA dataset.  Experiments are only based on a single dataset.

**Questions:**

Typo? The manuscript mentioned "Table 3.2," but this is not Table 3.2; I guess it should be Table 1.
Question: Will the medicine database be released to the public?

---

> ### Author Response · Authors · 2024-11-18
>
> Thanks for your constructive comments and suggestions for our manuscript. Below is our detailed response to address your concerns.
>
> # Limited data size
>
> Thank you for your valuable feedback. We hope the following clarifications address your concerns regarding the limited size of MedicineQA:
>
> 1. MedicineQA focuses on commonly used medicines to meet a wide range of prevalent needs. We aim for models performing well on MedicineQA to provide basic healthcare support in regions with relatively scarce medical resources. We began by collecting records from five major online consultation websites, which are rich sources of real patient-doctor dialogues. From such a large volume of data, we extracted conversations involving predefined commonly mentioned medicines. These dialogues were then utilized for evidence retrieval and dialogue revision, as detailed in the **Appendix**. Although MedicineQA contains only 300 samples, it covers most medicines commonly used in real-world scenarios across ten aspects of medicine application.
>
> 2. The construction of MedicineQA involved significant input from medical experts. Each step of the process was guided by real-world data and supervised by physicians to ensure the benchmark's quality. Our dataset construction was carried out by a panel of five board-certified physicians, with a senior advisor overseeing the entire process. Each sample in the benchmark, including the questions, search results, and answers, was carefully reviewed and revised by experts to ensure that the progression of questions was logical and the dialogue content accurately reflected real-world scenarios. Thus, we made a deliberate trade-off between scale and quality.
> 3. MedicineQA was specifically developed to address gaps in existing medical QA datasets by solving medication consultation scenarios with retrieval-augmented generation. Specifically, it relies on an authoritative database to provide relevant medicinal knowledge. The knowledge source for our RAG model is a subset of a private Chinese medicine database from an authoritative institution. We plan to expand the benchmark in the future to include more specialized medications. This expansion will require integrating additional medicines into our knowledge source. We are currently reviewing more medicines. The rigorous review and verification process is time-consuming, and we regret any delays in providing a larger benchmark. We appreciate your understanding and patience.
>
> # Details of Construction and Benchmark
>
> We apologize for any confusion caused by the omission of certain details regarding the construction process and the resulting benchmark. Specifically, after the physicians summarized the selected records into a question about a specific medicine, we prompted GPT-4 with the context of the relevant record to generate multi-round dialogues. This approach ensured that the generated dialogue content accurately reflected real-world scenarios. To prevent GPT-4 from hallucinating inappropriate content, physicians manually revised the dialogue to ensure it was realistic within the medical context and consistent with medical knowledge and logic. As a result, MedicineQA consists of 300 samples covering over 150 medicines, spanning 10 aspects (from Recommendation to Storage). In total, MedicineQA contains 989 rounds of dialogue.

---

> > ### Author Response · Authors · 2024-11-18
> >
> > # Details of Experimental settings
> >
> > We apologize for any confusion caused by the omission of details regarding the experimental settings. And we greatly appreciate your careful reading of our work. Our RagPULSE was trained on the general domain synthetic dataset but not for the medical domain. To create the synthetic dataset, we first collected a large-scale set of questions from multiple websites (e.g., Google and Baidu), including but not limited to dialogue questions and search engine queries. We then prompted GPT-4 to utilize [function call](https://platform.openai.com/docs/guides/function-calling) for summarizing the questions, resulting in the generation of a dialogue distilling dataset with 161,100 samples. To develop a model for the medication consultation scenario, we chose [PULSE](https://github.com/openmedlab/PULSE) as the baseline model, which shows notable performance in the medical domain. To enable PULSE to perform dialogue distillation, we add the synthetic dataset to the fine-tuning datasets of PULSE. This fine-tuning process took approximately **x** days using 8 NVIDIA A100 GPUs. We utilize the Adam optimizer with a weight decay setting of $0.01$ and betas of $(0.9, 0.95)$. The learning rate gradually decays from $9 \times 10^{-6}$ to $9 \times 10^{-7}$ following a cosine annealing learning rate schedule. Our training architecture integrates ZeRO-style data parallelism [1] with Megatron-style model parallelism [2]. We conducted both forward and backward computations in BFloat16, while optimizer updates were performed in Float32. Further details of the fine-tuning datasets can be found in the table below.
> >
> > | Category | Basic Q&A | Mathematical Abilities | Medical Q&A | Medical Exams | Synthetic dataset (Tool Call) | Model Safety | Multilingual |
> > | -------- | --------- | ---------------------- | ----------- | ------------- | ----------------------------- | ------------ | ------------ |
> > | Number   | 1015602   | 17364                  | 19055       | 338934        | 161100                        | 23421        | 313462       |
> >
> > [1] Rajbhandari S, Rasley J, Ruwase O, et al. Zero: Memory optimizations toward training trillion parameter models[C]//SC20: International Conference for High Performance Computing, Networking, Storage and Analysis. IEEE, 2020: 1-16.
> >
> > [2] Shoeybi M, Patwary M, Puri R, et al. Megatron-lm: Training multi-billion parameter language models using model parallelism[J]. arXiv preprint arXiv:1909.08053, 2019.
> >
> > # Details of our method
> >
> > We apologize for any misunderstanding caused by the omission of methodological details. To clarify, there is no dialogue distilling module in our proposed *Distill-Retrieve-Read* framework. The "Distill" step in our approach is, in fact, a straightforward and natural process. Summarizing medical dialogues is inherently more challenging than traditional dialogue summarization, as it involves extracting medically relevant information from multi-round conversations. Previous work has demonstrated that large language models (LLMs) outperform traditional language models in dialogue summarization tasks. In this work, our approach innovatively integrates the summarization process directly with the LLM's tool-calling capabilities rather than relying on an additional module to generate search queries. Specifically, we prompt the LLM to generate search keywords for several predefined search engines. When a query comes, the LLM mimics the process by which a human would use search engines, generating appropriate search keywords based on the current context of dialogue.  With the above paradigm, the LLM can call the search tool and generate the retrieval query according to the current dialogue.
> >
> > # Limited Evaluation
> >
> > We acknowledge your concern regarding the scope of our evaluation. Our primary focus is to address the unique challenges of medication consultation scenarios, which require an authoritative medicine database to provide relevant knowledge. To the best of our knowledge, existing datasets for medication consultation are not accompanied by such an annotated medicine database. Questions in the previous dataset lack precise, relevant medical documents. We develop MedicineQA to provide a rigorous benchmark for this purpose. In experiments, most of the LLMs we choose are medical LLMs, including PULSE, due to their relevance and the specificity of the domain. Thus, we did not conduct more experiments on other scenarios.

---

> > > ### Author Response · Authors · 2024-11-18
> > >
> > > # Q1:Typo
> > >
> > > Thanks very much for your careful reading. We are lucky to have met such a rigorous reviewer like you. What we intended to express was that the referenced content can be found in Section 3.2, Table 1. We appreciate your valuable feedback and will make the revisions in the updated version.
> > >
> > >
> > >
> > > # Q2: Publication of the medicine database
> > >
> > > We are very glad to release the medicine database to the public. The medicine database used in our paper is a subset of a Chinese medicine database from an authoritative institution. The whole database includes detailed descriptions of hundreds of thousands of medications, encompassing both traditional Chinese and Western medicines. We are currently in the process of translating and validating subsets of the database. We regret any delays in making this resource available as an open-source tool and appreciate your understanding. It will be available soon.

---

> > > > ### Author Response · Authors · 2024-11-25
> > > > **Sincerely looking forward to more discussion with you**
> > > >
> > > > Dear Reviewer rJMn,
> > > >
> > > > Thank you for the precious review time and insightful comments on our paper. We have provided detailed responses and results. Specifically, we have:
> > > >
> > > > 1. **Added more details** in the methods and experimental sections to improve clarity
> > > > 2. **Clarified the scale of the benchmark** and included a detailed explanation in the revised manuscript.
> > > >
> > > > If you could share your thoughts on the updates or let us know if there are any remaining concerns that we could further address, we are more than happy to provide additional clarification. Sincerely look forward to your reply!
> > > >
> > > > Best,
> > > > Authors

---

> > > > > ### Comment · Reviewer_rJMn · 2024-11-25
> > > > > **Score is raised and the details need to be updated in the revised version**
> > > > >
> > > > > Given the details provided in the response, and the promise that the data will be released, I have raised my score.

---

> > > > > > ### Author Response · Authors · 2024-11-25
> > > > > >
> > > > > > We sincerely appreciate your thorough review and constructive feedback on our manuscript. Your detailed comments have provided invaluable insights that have significantly contributed to the refinement of our work.

---

> ### Author Response · Authors · 2024-12-04
>
> Dear Reviewer rJMn,
>
> We would like to sincerely thank you for your time and effort in engaging during the discussion phase. We greatly appreciate the constructive feedback and thoughtful comments you provided, which have been invaluable in improving the quality of our work.
>
> Best regards,
>
> Authors

---

### Author Response · Authors · 2024-11-18
**Motivation of Our Paper**

We sincerely appreciate all reviewers’ time and effort in reviewing our paper. We are pleased that the reviewers generally appreciate MedicineQA. We are willing to release a subset of the medicine database to support research and applications. Currently, we are in the process of translating and validating subsets of the database. We regret any delays in making this resource available as an open-source tool and appreciate your understanding.

# Motivation

We hope the following clarification will provide a more comprehensive understanding of our work. In recent years, large language models (LLMs) have demonstrated impressive capabilities and can be repurposed with minimal effort across numerous domains and diverse tasks. These models have shown great promise in medicine, with applications including primary care, clinical decision support, and medication consultation. However, the bar for clinical applications is particularly high due to the safety-critical nature of the medical domain. This is especially important for LLMs, as they are prone to hallucinating convincing but potentially harmful medical misinformation. Previous works sought to incorporate relevant evidence via Retrieval Augmented Generation. The paradigm of retrieving external knowledge holds great promise for knowledge-intensive tasks like medication consultation. Nevertheless, attempts to assess the capabilities of RAG-based LLMs in medical scenarios are limited. To address this gap, we introduce MedicineQA, a benchmark comprising 300 high-quality, expert-annotated multi-round dialogues spanning ten key aspects of medication consultation that users commonly encounter on online consultation platforms.

MedicineQA focuses on commonly used medicines to meet a wide range of prevalent needs. Our goal is for models that perform well on MedicineQA to provide basic healthcare support in regions with relatively scarce medical resources. Besides, we harness the trivial RAG-based LLMs with a novel *Distill-Retrieve-Read* framework to access the external knowledge. The model processes the inquiry by summarizing the dialogue history into keywords, mimicking the process of calling search engines, and then integrating the retrieved evidence to formulate a comprehensive response.

# Revised Paper

According to the reviewers' comments, we have revised our paper and marked the modified parts in blue for clarity.

---

### Author Response · Authors · 2024-11-29

Dear Reviewers,

We would like to extend our heartfelt gratitude for your valuable and constructive feedback during the previous round of discussions. Your insights have been instrumental in refining and improving our revised manuscript.

In this paper, we present MedicineQA, a dataset centered on commonly used medicines, designed to address a wide range of prevalent healthcare needs. Our goal is for models excelling on MedicineQA to provide basic healthcare support in regions with limited medical resources. Additionally, we are committed to offering proofreading and translation services for a subset of our medicine database to ensure comprehensive and accessible support. We hope our work will inspire further innovation in applying large language models (LLMs) to the medical domain and help bridge the gap between these models and real-world applications.

Given the extended timeline, we humbly and sincerely request the opportunity to continue our dialogue with you to fully meet your expectations. If possible, we kindly ask you to reassess our revised manuscript. If the updates adequately address your concerns, we would be deeply grateful if you would consider revisiting and potentially revising your score. If there are still unresolved issues or specific areas where the manuscript could be further improved, we would greatly appreciate any additional feedback. Your input remains invaluable in helping us achieve the highest standard for this work.

Thank you once again for your time, thoughtful feedback, and continued engagement.

Best regards,

The Authors

---

### Note · Authors · 2025-01-22

I have read and agree with the venue's withdrawal policy on behalf of myself and my co-authors.